# Deconstructing Sox2 Function in Brain Development and Disease

**DOI:** 10.3390/cells11101604

**Published:** 2022-05-10

**Authors:** Sara Mercurio, Linda Serra, Miriam Pagin, Silvia K. Nicolis

**Affiliations:** Department of Biotechnology and Biosciences, University of Milano-Bicocca, 20126 Milan, Italy; lindaserra91@gmail.com (L.S.); miriam.pagin@unimib.it (M.P.); silvia.nicolis@unimib.it (S.K.N.)

**Keywords:** Sox2, neural stem cells, neurons, glia, development, brain, transcription factor

## Abstract

SOX2 is a transcription factor conserved throughout vertebrate evolution, whose expression marks the central nervous system from the earliest developmental stages. In humans, *SOX2* mutation leads to a spectrum of CNS defects, including vision and hippocampus impairments, intellectual disability, and motor control problems. Here, we review how conditional *Sox2* knockout (cKO) in mouse with different Cre recombinases leads to very diverse phenotypes in different regions of the developing and postnatal brain. Surprisingly, despite the widespread expression of *Sox2* in neural stem/progenitor cells of the developing neural tube, some regions (hippocampus, ventral forebrain) appear much more vulnerable than others to *Sox2* deletion. Furthermore, the stage of *Sox2* deletion is also a critical determinant of the resulting defects, pointing to a stage-specificity of SOX2 function. Finally, cKOs illuminate the importance of SOX2 function in different cell types according to the different affected brain regions (neural precursors, GABAergic interneurons, glutamatergic projection neurons, Bergmann glia). We also review human genetics data regarding the brain defects identified in patients carrying mutations within human *SOX2* and examine the parallels with mouse mutants. Functional genomics approaches have started to identify SOX2 molecular targets, and their relevance for SOX2 function in brain development and disease will be discussed.

## 1. Introduction

SOX2 is an HMG-box containing transcription factor belonging to the SOXB1 subgroup of *Sox* genes (which also includes *Sox1* and *Sox3*) [1]. It is essential for embryonic development from the first stages of embryo formation to the development of multiple areas of the nervous system. It is expressed in the inner cell mass of the mouse embryo, which will give rise to the embryo proper; its role is so important that its ablation by a homozygous null mutation results in the arrest of embryonic development, just after the embryo implants in the uterus [2]. Later in development, *Sox2* expression marks the forming nervous system, from neural induction onwards; here, *Sox2* is expressed throughout the neuroepithelium (undifferentiated stem-progenitor cells), forming the developing neural tube and also in some neurons and glia [3]. Given its essential role in maintaining the pluripotent stem cells of the early embryo, SOX2 has been further investigated (together with other genes) in respect of its potential to recreate pluripotency in differentiated cells and has indeed been identified as one of the four factors able to reprogram differentiated somatic cells to pluripotency, thereby generating induced pluripotent stem cells (iPSC) [4,5].

To study its role in the development of the nervous system, conditional knock-outs (cKO) were generated to ablate *Sox2* in different areas of the developing nervous system and at different developmental time points. Even though *Sox2* is broadly expressed in the nervous system, these experiments show that it is essential for the development of only specific areas of the brain.

In this review, we will discuss insights provided by *Sox2* cKO models (Table 1) that highlighted specific SOX2 functions in the developing telencephalon (hippocampus, medial ganglionic eminences), diencephalon (visual thalamus, hypothalamus), and cerebellum. We will summarize evidence that SOX2 controls the development of different brain areas acting at the level of neural stem/progenitor cells, but also of specific differentiated neuronal and glial cell types. We will review findings describing how SOX2 controls specific target genes in the context of different neural cell types and brain regions and discuss evidence that some of them represent mediators of Sox2 function in neurodevelopment. We will discuss studies on ex-vivo brain-derived neural stem cell (NSC) cultures that have been instrumental in identifying genes directly regulated by SOX2, in demonstrating their role as effectors of SOX2 function in NSC self-renewal and neuronal/glial differentiation, and in uncovering a novel molecular mode of action for SOX2 in gene regulation. Finally, we will also discuss some implications of mouse models of SOX2 function for the understanding of human genetic disease, caused by heterozygous mutations in *SOX2*.

SOX2′s role in sensory organs and pituitary gland will not be addressed here and can be found elsewhere [3,6,7,8,9,10].

## 2. SOX2 Function in Different Brain Regions

### 2.1. Telencephalon

*Sox2* is expressed in the developing telencephalon in neural progenitors, both ventrally and dorsally. To assess its role in the formation of the telencephalon, cKO mice have been generated by employing different Cre recombinases to assess its function at different developmental time points and in different telencephalic regions. Two telencephalic areas have been shown to require a functional SOX2 for their development: the hippocampus (medio-dorsal telencephalon) and the ganglionic eminence (GE) (ventral telencephalon).

#### 2.1.1. Hippocampus

The hippocampus is a part of the brain responsible for memory formation and one of the brain regions where neurogenesis persists throughout life, via NSC of the dentate gyrus (DG) [20,21]. It forms in the posterior-medial telencephalon through a series of complex reorganization events. Hippocampal formation initially requires signals from the cortical hem (CH), positioned in the medial telencephalic wall, clearly distinguishable in mice at E12.5 (Figure 1A), and able to organize surrounding tissues into a hippocampus [20,22,23]. NSC and intermediate neural progenitors (IP) migrate from the dentate neural epithelium (DNE), adjacent to the CH, along glial fibers to eventually organize the DG. Cajal-Retzius cells, derived from the CH, have a key role in DG formation [21,24,25] (Figure 1A).

*Sox2* is expressed in the developing hippocampus from the beginning of its development, in the CH, and in the DNE, and then remains expressed in the DG throughout life. *Sox2* expression is enriched in the CH, compared to the surrounding tissues [12,26], suggesting a key role for SOX2 in this region. *Sox2* is expressed by neural progenitors and then turned off when they differentiate into neurons [27]. A first evidence that SOX2 could be involved in hippocampal formation came from the analyses of hypomorphic *Sox2* mutant mice (hypo Sox2 KO), compound heterozygous for a null *Sox2* allele (on one chromosome) and a deletion of a telencephalic enhancer (on the other chromosome); these mice presented a reduction in NSC in the adult dentate gyrus and a reduction in their ability to produce differentiated neurons [26]. A more severe postnatal defect in hippocampal formation was observed by conditionally ablating *Sox2* in neuroepithelial (neural stem/progenitor) cells by crossing a Sox2^flox^ allele with a Nestin-Cre transgene (Sox2-Nestin-Cre cKO), in which Cre activity is driven by an enhancer of the *Nestin* gene [28]. In these mice, the DG appeared unaffected at birth, but apoptosis of hippocampal NSC was observed after birth, leading to a DG with reduced numbers of stem cells about a week after birth [15] (Figure 1A). Because *Sox2* deletion in these mice occurs early during development (starting at E11.5), the postnatal defects observed could be due to a developmental role of SOX2; however, even *Sox2* deletion in the adult hippocampus results in reduction of NSC and granule neurons, pointing to a requirement of SOX2 throughout life for NSC maintenance [15]. The expression of key signaling molecules during embryogenesis of the hippocampus is affected by *Sox2* loss. In fact, *Shh* and *Wnt3A* expression is downregulated; reactivating SHH signaling in *Sox2* mutants via a pharmacological agonist, or in *Sox2* mutants NSC in culture, rescues the proliferation defects, pointing to SHH signaling as a functional contributor to SOX2-dependent NSC maintenance [15]. *Sox2* mutant NSC reduction, both in vivo and in vitro, is preceded by an increase in apoptosis; this could be linked to the finding that SOX2 regulates the expression of Survivin, a known inhibitor of cell death, in NSC in culture [15,29].

Nestin-Cre is expressed in the neural tube starting at E11.5 in Sox2-Nestin-Cre cKO; by E12.5, *Sox2* is deleted in the dentate neural epithelium, and by E14.5 in the cortical hem [12,15] (Table 1). What would happen if Sox2 was to be ablated even earlier in development in the telencephalon?

Recently, cKO of *Sox2* in the developing telencephalon at time points earlier than Nestin-Cre-mediated *Sox2* deletion has been shown to cause defects much more severe than what was observed in Sox2-Nestin-Cre cKO, affecting hippocampal development before birth [12] (Figure 1A). Two different Cre lines were crossed to the Sox2^flox^ allele: FoxG1-Cre, active at E9.5, and Emx1-Cre, active at E10.5, to generate a Sox2-FoxG1-Cre cKO (early Sox2 cKO) and a Sox2-Emx1-Cre cKO (intermediate Sox2 cKO), respectively. In the early Sox2 cKO, the DG was completely missing just before birth, the hippocampal glial scaffold was disorganized, and Cajal-Retzius cells, important in organizing hippocampal morphogenesis, were greatly reduced in number. In the Sox2-Emx1-Cre cKO, the DG was much more affected than in Sox2-Nestin-Cre cKO, but not as much as in Sox2-FoxG1-Cre cKO, pointing to different requirements for Sox2 at different developmental time points [12] (Figure 1A).

To identify key SOX2 targets, whose expression could be affected by *Sox2* loss and lead to the observed defects, the expression of genes directly regulated by SOX2, previously identified in NSC by SOX2 ChIPseq and RNAseq experiments [30], was analyzed in vivo. In particular, the expression of genes already known for their involvement in hippocampus development were studied.

SOX2 binding sites were found in an intron of the gene encoding the GLI3 transcription factor in NSC, suggesting that *Gli3* could be a direct SOX2 target, and indeed *Gli3* expression was found to be downregulated in the FoxG1-Cre cKO. Interestingly, *Gli3* expression was not affected when *Sox2* was ablated later in development, as in the Sox2-Emx1-Cre cKO or Sox2-Nestin-Cre cKO; therefore, its downregulation might account for the stronger phenotype observed in the FoxG1-Cre cKO [12]. Additionally *Wnt3A* expression is almost absent in the early *Sox2* mutant, even though it does not appear to be a direct target in NSC; interactions between Sox genes and the Wnt pathway have been described previously [31]. Rescue of the hippocampal phenotype in vivo by administering a Wnt agonist to the mice was attempted, but it did not show a complete rescue of the phenotype, likely due interference with embryo survival. Administering Wnt agonist to *Sox2* mutant NSC in culture could avoid mortality problems seen in vivo and might elucidate the role of Wnt signaling in Sox2 regulation of forebrain NSC proliferation and survival.

#### 2.1.2. Medial Ganglionic Eminence (MGE)

Gamma-aminobutyric acid-containing (GABAergic) cortical interneurons (CIN) are inhibitory neurons in the cerebral cortex, and they are essential in regulating communication between cortical neurons [32,33]. They are generated during development mainly from the proliferative areas of subpallial regions of the telencephalon, which include the medial ganglionic eminence (MGE), the caudal ganglionic eminence (CGE), and the preoptic area (PoA). Once they become postmitotic, they migrate tangentially to the forming cortex and then migrate radially to position themselves in the forming cortical layers [32,34].

*Sox2* is expressed in the proliferative zone of subpallial regions of the telencephalon where GABAergic CIN originates [26], and a hint that Sox2 could be involved in their differentiation came from the observation that NSC from the forebrain of hypo *Sox2* KO (see above) are deficient in the differentiation into GABAergic CIN in vitro. In addition, even in vivo, fewer GABAergic CIN are found in the adult cortex of hypomorphic Sox2 mice, and they show morphological abnormalities [35]. The reduced numbers of CIN in the mutant cortex is likely due to an impairment in their migration from the MGE [35], while their abnormal morphology could be due to the fact that Sox2 mutant neurons co-express glial and neuronal markers. SOX2 was shown to promote neuronal differentiation by inhibiting the glial marker GFAP; this does not happen in *Sox2* mutant neurons, and this “confused” state could impair proper neuronal differentiation [35]. Interestingly, *Sox2* is expressed in all NSC, but only one kind of neuron seems to be specifically affected, GABAergic CIN.

The requirement for SOX2 in MGE development came from analyses of the Sox2-FoxG1-Cre cKO in which the MGE was severely impaired following *Sox2* ablation from E9.5 in the whole telencephalon [36] (Figure 1B, Table 1). MGE loss (as seen by the disappearance of markers such as Nkx2.1 and Shh, two direct SOX2 targets, and by morphology) was preceded, even in this mutant, by increased cell death in the ventral telencephalon, pointing to a role of Sox2 in regulating cell death, as mentioned for hippocampal development (see above) [36]. Interestingly, *Sox2* ablation a little later in development, in Sox2-Nestin-Cre cKO, lead to a much milder phenotype, suggesting a requirement for Sox2 in MGE development between E9.5 and E11.5 [15]. Having a defective MGE results in a reduction in the production of GABAergic CIN; indeed, CIN are greatly reduced in the cortex of Sox2-FoxG1-Cre cKO [36]. Which SOX2 targets could mediate its role in MGE maintenance?

Two genes important for MGE development are downregulated in Sox2-FoxG1-Cre cKO, encoding the secreted molecule SHH and the homeobox transcription factor NKX2.1. Shh had already been shown to be downstream of Sox2 in DG development, and reactivation of the SHH pathway in Nestin-Cre-Sox2 cKO mutants rescues the DG phenotype in vivo and the proliferation of NSC in vitro [15] (Figure 1B). Similarly, administration of a SHH agonist to Sox2-FoxG1-Cre cKO rescues the MGE phenotype, pointing to SHH as a mediator of SOX2 function in different telencephalic domains. The *Nkx2.1* gene is a direct SOX2 target, and the NKX2.1 transcription factor is known to directly regulate *Shh* expression [37,38], leaving open the possibility that SOX2 regulation of Shh could be both direct [15] and via NKX2.1.

### 2.2. Diencephalon

SOX2 is known as the “stem cell” factor required for the maintenance of embryonic and NSC; however, recently its role in differentiated neurons and glia has been described (reviewed in [3]). Indeed, in the diencephalon, *Sox2* has been found expressed in differentiated neurons in multiple nuclei [19,39,40,41]; in this review, we will discuss its role in the neurons of the dorso lateral geniculate nucleus (dLGN) in the thalamus and of the suprachiasmatic nucleus in the hypothalamus.

#### 2.2.1. Thalamus

The thalamus is a brain district that connects sensory organs to the cerebral cortex. It includes nuclei specialized in receiving and processing sensory information: the lateral geniculate nucleus (dLGN) receives visual stimuli from the retina and projects to the visual cortex; the ventral posterior nucleus (VPN) receives somatosensory stimuli from the periphery/skin and projects to the somatosensory cortex, and the medial geniculate nucleus (MGN) receives auditory information from the ear and projects to the auditory cortex.

*Sox2* is expressed in the retina [42,43,44], the ear [45], and the skin [46], in neural progenitors of the diencephalon, and also in differentiated neurons in the three thalamic nuclei that receive and process sensory information (Allen Brain Atlas and [19]). While SOX2′s role in sensory organs has received attention in the past years, until recently very little was known on the role of SOX2 in the developing thalamus [19].

Previously, a reduced size of the thalamus was observed in the hypo *Sox2* KO [26], but only recently *Sox2* was ablated specifically in differentiated neurons of dLGN, VPN, and MGN during development, by crossing the Sox2^flox^ allele with a Roralpha-Cre transgene (Sox2-Rora-Cre cKO) specifically expressed in dLGN, VPN, and MGN from E14.5 [18,19].

Even though *Sox2* is ablated in all three nuclei, there is a difference in the phenotype observed: the dLGN is the most affected, followed by the VPN, while the MGN does not appear strongly reduced in size [19]. Interestingly, the dLGN is the thalamic nucleus with the strongest *Sox2* expression [19] and is therefore probably sensitive to SOX2 levels. Indeed, the formation of the visual system is greatly affected. At birth, a reduction of retinal fibers reaching the dLGN is observed in the Sox2-Rora-Cre cKO, retinal fibers do not segregate correctly in the mutant dLGN at P7, and the mutant dLGN is reduced in size. Projections from the thalamus to the primary visual cortex (V1) are greatly reduced in *Sox2* thalamic mutants compared to controls. In order for V1 to differentiate, dLGN-cortical projections need to be properly established, otherwise primary (V1) and secondary visual areas do not develop correctly [18]. In fact, the V1 of Sox2-Rora-Cre cKO is not correctly patterned [19] (Figure 2A).

In a search for SOX2 targets that could mediate its function in the formation of the visual system, two signaling pathways, important for correct pathfinding in the brain, were found affected in Sox2-Rora-Cre cKO: Eph/ephrin signaling and Serotonin signaling. Ephrin-A5 (EFNA5) is expressed in a gradient in the dLGN, and it is important, together with EFNA2 and EFNA3, in guiding retinal axons to the dLGN [47,48]. In *Sox2* thalamic cKO, *Efna5* is specifically downregulated in the dLGN [19]. In addition, we identified a region, within the *Efna5* locus that is bound and activated by SOX2, likely acting as an *Efna5* enhancer. It is therefore likely that *Efna5* downregulation could be one of the reasons for the altered retinogeniculate projections in the *Sox2* mutants [19].

In addition to Ephrin/Eph signaling, Serotonin signaling has been shown to be important for retinogeniculate projections [49,50]. Levels of Serotonin and its transporters SERT and vMAT are found downregulated in the dLGN of *Sox2* mutants, suggesting a possible involvement of serotonergic signaling in visual system development downstream of SOX2 [19].

#### 2.2.2. Hypothalamus

*Sox2* is co-expressed with neuronal markers, not only in the thalamus, but also in the hypothalamus. It is expressed in cells positive for NeuN (a marker of differentiated neurons) in the arcuate (ARC) and in the suprachiasmatic nuclei (SCN) [40,41], brain regions involved in homeostasis of food intake, and regulation of circadian rhythms, respectively.

In the ARC, *Sox2* is expressed in tanycytes, specialized ependymal cells with characteristics of NSC in the adult brain, and in cells expressing markers of differentiated neurons, such as NeuN [51]. Interestingly, SOX2 expression in NeuN-expressing cells is lost with aging and with obesity induced diets; SOX2′s role in this cell type deserves investigation [41]. In the mediobasal hypothalamus, which includes the ARC, *Sox2* is expressed in NSC that have been shown to control aging in part by the release of exosomal miRNAs in the cerebrospinal fluid [52].

Suprachiasmatic nuclei (SCN), paired structures in the anterior hypothalamus above the optic chiasm, are considered the central circadian pacemaker in mammals [40]. They receive visual stimuli for light or dark through indirect or direct retina-SCN pathways and generate circadian rhythms. Different clock genes have been cloned that are involved in a series of transcription–translation feedback loops (TTFL) that makes up the molecular clock (reviewed in [53]). Among the key genes involved in the molecular clock are Period1 (*Per1*) and Period2 (*Per2*).

Most PER2+ cells in the adult SCN express SOX2 [39]. To understand how SOX2 loss could affect the activity of SCN neurons, a cKO was generated in which *Sox2* was deleted in all GABA-ergic interneurons (using a vescicular GABA transporter (VGAT) Cre recombinase). While the structural organization of the SCN was not affected by *Sox2* conditional ablation, the expression of *Per2* and of neuropeptide genes was greatly reduced. These mice have a deficit in light-induced entrainment and display widespread changes in behavioral rhythms [39]. SOX2 was found to regulate the clock activity of SCN neurons by directly activating, in vitro and in vivo, the *Per2* gene. In addition, RNAseq experiments identified a reduction of transcription of other clock genes and also of neuropeptides. Therefore, SOX2 is thought to regulate signaling within the SCN nucleus and between the SCN nucleus and other parts of the brain [39] (Figure 2B).

### 2.3. Cerebellum

The cerebellum forms from rhombomere 1 in the hindbrain and is essential for movement; indeed, defects in cerebellum formation in humans lead to motor control defects and ataxia [54,55]. Early in development, the expression of two transcription factors defines the border between the midbrain and the hindbrain: *Otx2* is expressed in the forebrain and midbrain, while *Gbx2* is expressed in the hindbrain. The Otx2-Gbx2 boundary will become the isthmic organizer, an important signaling center. The correct expression of these transcription factors is required for the correct development of the cerebellum [56,57]. Postnatally, cerebellar neurons and glia continue their differentiation and interact to obtain a functional cerebellum.

*Sox2* is expressed in the neuroepithelium of midbrain and hindbrain, and later it is expressed in specific glial populations that include parenchymal astrocytes of the granular layer (GL) and prospective white matter (PWM), and Bergmann glia (BG), while it does not appear to be expressed in Purkinje neurons or other neurons [16]. However, BG does regulate neuronal activity since they surround with their cell bodies Purkinje cells and are critical for their function [58,59,60]. *Sox2* expression in BG has been found also in the human cerebellum [61].

Two different cKO mice have been generated to study SOX2 function in the cerebellum. In the first cKO, *Sox2* is ablated in midbrain and hindbrain neuroepithelium by E9.5 (by means of Wnt1-Cre, Sox2-Wnt1-Cre cKO), while the second is an inducible cKO in which *Sox2* is deleted postnatally through activation of a glial-specific Cre recombinase (Sox2-GLAST-CreERT2 cKO) (Figure 2C). The early *Sox2* deletion leads to an expansion of the *Otx2*-expressing domain into the hindbrain, a reduction of the cerebellar vermis, and ataxia in the mutants [16]. Similar phenotypes have been seen when *Otx2* is ectopically expressed in the hindbrain [57]. Interestingly, *Otx2* and Gbx2 are direct targets of SOX2 in forebrain-derived NSC [30], and potentially also in other parts of the CNS, such as the midbrain and hindbrain. In addition to the morphological defects described, a particular cell type is affected in the Sox2-Wnt1-Cre cKO postnatally, BG. BG is usually localized to the Purkinje cell layer, but in the *Sox2* mutants not only is it misplaced to the molecular layer, but its morphology is aberrant (Figure 2C). This BG defect is observed about 3 weeks postnatally, and it appears to be due to a cell autonomous requirement of SOX2 in this cell type, rather than to a developmental problem in cerebellum morphogenesis; in fact, even postnatal *Sox2* ablation, specifically in glia, in Sox2-GLAST-CreERT2 cKO, leads to misplacement and aberrant morphology of BG [16]. Movement defects are also observed when *Sox2* is deleted in BG postnatally, in the Sox2-GLAST-CreERT2 cKO; however, they are much milder compared to *Sox2* deletion in early development. This observation suggests that BG anomalies lead to movement defects in these animals, but these defects are made worse by vermis hypoplasia.

How could BG defects in the *Sox2* mutants lead to ataxia? It is known that BG is important in removing neurotransmitters from the synaptic cleft, and defects in this uptake could result in alterations of signals by PC. In fact, *Sox2* mutant BG are defective in glutamate uptake and synaptic transmission between parallel fibers and PC is altered [16].

## 3. Sox2 in Neural Stem Cells

*Sox2* is expressed in NSC and is essential for their self-renewal and differentiation into neurons [15,62]. Its expression is fine-tuned, both at the transcriptional and the post-translational level in order to have the right amount of SOX2 for correct neural development [63,64,65,66].

Analyses of mouse models have provided key information on how SOX2 functions in NSC.

*Sox2* cKO in NSC via Nestin-Cre (Sox2-Nestin-Cre cKO) leads to depletion of NSC in the hippocampus in vivo; in addition, ex-vivo NSC cultures, derived from forebrains dissected at birth, need SOX2 to continue dividing in culture; without SOX2, they are exhausted after a few passages. A key signaling pathway downstream of SOX2 required for NSC self-renewal is the SHH pathway; indeed, treatment of *Sox2* cKO NSC in culture with a SHH agonist rescued their proliferation defect [15]. Recently, multiple genome-wide analyses performed on NSC derived from Sox2-Nestin-Cre cKOs and control mice [15] allowed to add many new effectors to the gene regulatory network downstream of SOX2. Long-range interactions between gene promoters and distant regulatory elements have been identified in NSC chromatin by the ChIA-PET method, and many of these interactions are lost in *Sox2*-mutant NSC. In addition, genes downregulated in *Sox2* mutant NSC are involved in long-range interactions, and their distant enhancers are highly enriched in SOX2 binding in WT NSC [30]. Among the genes whose expression is downregulated in *Sox2* mutant NSC are transcription factors and signaling molecules; these include *Socs3*, encoding an inhibitor of JAK/STAT signaling, and *Fos* and *Jun*, whose products make up the AP1 complex important for different cellular processes including proliferation, differentiation, and apoptosis [67] (Figure 3).

*Socs3* is strongly downregulated in *Sox2* cKO NSCs; it is directly bound by SOX2 on the promoter, is involved in multiple interactions, and its overexpression in *Sox2* mutant NSCs rescues the proliferation phenotype. Another direct SOX2 target able to rescue the proliferation defect is *Fos*; the *Fos* gene product is also able to directly activate *Socs3* [68]. Therefore, in addition to the SHH pathway, a SOCS3/FOS regulatory loop downstream of SOX2 is important for NSC maintenance (Figure 3). Whether the Shh pathway and Socs3/Fos interact is not yet known.

*Sox2* mutant NSCs are deficient, not only in the ability to self-renew, but also in differentiating into neurons; overexpression of *Fos* (but not *Socs3*) is able to rescue this defect (Figure 3B) [68,69].

Interestingly, CUT&RUN studies show that, genome-wide, SOX2, JUN, and FOS (AP1 complex) bind together to DNA; in particular, genes expressed and important in NSCs, and also important for neuronal differentiation (such as *Socs3*), are downregulated in *Sox2* mutant NSCs, and directly bound by SOX2 and the AP1 complex (Figure 3) [69]. Interestingly, these studies document that a new gene regulatory loop involving SOX2, SOCS3, and FOS may be required for NSCs proliferation and, later, differentiation into neurons.

These recent studies on the role of SOX2 in NSC have revealed a previously unknown role of SOX2 in mediating long-range interactions in the chromatin. These studies also identified regulatory regions, bound by SOX2, connected to genes known to be involved in neurological disorders; these could be novel sites of mutations linked to such diseases [70]. It is important to remember that a given enhancer is very often not connected to the nearest promoter, but rather to a more distant one(s), skipping genes in between; further, an enhancer can be located within the intron of a gene, and be connected to the promoter of a different gene [30,71]. Hence, long-range interaction maps are critical to annotate a given enhancer to the correct gene. In the case of mutations affecting an enhancer, interaction maps can improve our prediction of which gene may alter its expression as a consequence of the mutation. Indeed, most of the enhancers and promoters involved in the identified mouse long-range interactions are conserved in humans, in syntenic chromosome regions. Interestingly, sequence variants, identified by genome-wide association studies (GWAS) to be associated with neurodevelopmental diseases and traits (schizophrenia; bipolar disorder; intelligence), are found to be localized within these human enhancers, suggesting that they could contribute to the disease by the deregulation of the connected gene (D’Aurizio et al., submitted). In many cases, the variant associated in humans to a neurodevelopmental disease affects an enhancer whose mouse counterpart interacts with the mouse homolog of a gene already associated with a human disease, strengthening the hypothesis that the gene is involved in the disease. Other enhancers are connected to genes not previously associated with the disease, pointing to their possible pathogenetic involvement. Extending these studies to newly discovered variants may in the future allow the identification of new disease genes (D’Aurizio et al., submitted).

## 4. Specific Partnerships as a Basis for SOX2 Cell-Type-Specific Functions in the CNS?

Based on the studies reviewed above, SOX2 appears to play region-specific and cell-type specific functions within the developing CNS, rather than “general” neural functions. What molecular mechanisms may be at the basis of this? A possible hypothesis resides in partnerships of SOX2 with other transcription factors. Seminal work by the Kondoh laboratory proved the central role of the synergistic interaction between SOX2 and PAX6 in the binding to crystallin gene enhancers and its essentiality for gene activation and lens development, and extended the proposal of the relevance of specific partnerships to the function of SOX proteins in general [72,73]. SOX2 physically and functionally interacts, in embryonic stem (ES) cells, with NANOG to regulate ES cell self-renewal [74]; the importance of partnership in SOX2 function is further emphasized by the observation that SOX2 co-occupies enhancers in combination with different POU factors in ES cells (OCT4/Pou5f1) and NSC (BRN2/Pou3f2), and that the ectopic expression of the transcription factor BRN2 in mouse ES cells caused a genome-wide re-location of SOX2 binding to a neural set of target sites, with the acquisition of aspects of a neural differentiation program [75]. In NSC, SOX2 co-binds to DNA in partnership with CHD7 (in NS-5 NSC, [76], and AP1 (in forebrain-derived NSC, [68,69], see above), acting on the regulation of targets important for key functions such as NSC self-renewal and differentiation to GABAergic neurons. SOX2 ChIPseq experiments have been carried out on primary tissues including mouse cerebral cortex, spinal cord, stomach, and lung/esophagus; SOX2 bound to a related motif in these different cell types; however, its target binding sites were found to be remarkably cell-type-specific, and they were enriched for the binding motifs of interacting cofactors, suggesting again that specific partnerships constitute an important element guiding SOX2 binding in different cell types [77]. Given the relevance of SOX2 in shaping the 3D promoter-enhancer long-range interaction network in NSC [30,70], it is possible that its interaction with different partners provides a mechanism for the formation of different interaction loops in different cell types, and/or that SOX2 “seeds” the chromatin by the formation of loops involving “poised” enhancers, which are then further regulated to become fully active enhancers (or silencers) by the interaction of SOX2 with different partners. The investigation of SOX2 partners in SOX2-dependent cell types (such as thalamic projection neurons; Bergmann glia, etc.) may provide a fertile perspective to better understand the basis of the specificity of SOX2 functions in the CNS.

The capacity of SOX2 to mediate long-range interactions in the chromatin is probably required, not only for NSC maintenance, but also for its role in the reprogramming of somatic cells into iPSCs. Indeed, SOX2′s role in chromatin opening has been shown to be instrumental for pluripotency establishment [78]. Comparison of SOX2 levels and partners in iPSC, NSC, and ESC will help elucidate the different mechanisms of stemness in the future.

## 5. *SOX2* Dysfunction in Human Disease: Insights from Mouse Models

Heterozygote loss of function mutations in the *SOX2* gene leads to rare human disorders characterized by anophthalmia, lack of one or both eyes, or microphthalmia, unusually small eyes [79,80]. In addition, *SOX2* mutations lead to defects in the function of different brain areas, resulting in seizures, hypoplasia of the hippocampus, cognitive impairment, and motor defects [81].

*Sox2* expression in the brain is greatly conserved between mouse and humans [9,10,82], and the mouse models previously described have identified some cellular abnormalities possibly contributing to the human phenotypes. Electrophysiological studies of the hippocampus lacking *Sox2* in mouse identified an altered excitatory activity in CA1 and CA3 [12], and it is possible that this contributes to the observed seizures in humans. Hypoplastic hippocampi have been found in *Sox2* mouse mutants, plausibly due to depletion of stem cells, mirroring the hippocampal hypoplasia observed in *SOX2*-mutant patients by magnetic resonance imaging (MRI) [12,15,80]. Hippocampal hypoplasia is known to cause seizures and is thus a plausible contributor to the epilepsy-like pathology observed in *SOX2*-mutant patients [80]. Furthermore, postnatal hippocampal neurogenesis by NSC has been shown to contribute to learning and memory in mouse models [83,84], and its defects may contribute to the intellectual disability found in most *SOX2*-deficient patients.

Motor defects have been reproduced in mice lacking *Sox2* in the cerebellum. In these mice, a key portion of the cerebellum, the vermis, is reduced, but also a cell type in which *Sox2* is expressed, Bergmann glia (BG), is disorganized, mislocalized, and reduced in number [16]. Similar defects in BG have been described in Vanishing white matter leukoencephalopathy, a disease characterized by motor defects among other deficits [85], and a link between vermis hypoplasia and ataxia is known [86]. These observations suggest that the defects observed in *Sox2* KO mice could be present in humans carrying *SOX2* mutations.

The studies of SOX2 function in mice can not only help in understanding the bases of the human disease caused by *SOX2* mutation, but can also point to previously unrecognized defects in humans that could be due to mutations in the *SOX2* gene. In fact, *Sox2* ablation in the visual thalamus in mouse resulted in problems with the differentiation of the visual system, defects found in humans with cerebral visual impairment [19]. It will be interesting in the future to search for mutations in *SOX2*, *SOX2* regulatory elements, or SOX2 target genes in patients with cerebral visual impairment.

## 6. A “Dark Side” for SOX2 Function in Neural Disease: Maintenance of Tumor-Initiating Cells in Gliomas

In parallel with the emerging function of SOX2 in various types of stem cells, and in particular in neural stem cells (see above, Section 3), the concept of “cancer stem cells” (CSC) or “tumor-initiating cells” (TIC) was emerging from studies on the cellular basis of tumorigenesis: tumors were found to be heterogeneous, and to contain a fraction of cells able to re-initiate tumor development following transplantation; these cells were resistant to most conventional chemotherapy, and were responsible for tumor relapse. Neural tumors in particular were among the first tumors in which CSC/TIC were demonstrated, and these cells were shown to express SOX2. These observations prompted the functional evaluation of SOX2 roles within these cells, and these studies led to the demonstration of an essential role for SOX2 in the maintenance of cancer stem cells within gliomas and glioblastomas (the most severe and deadly neural tumors), as well as in CSC/TIC in several more common tumor types; this functional relevance was proven in both patient-derived cells and mouse models. SOX2 expression and function in neural CSC/TIC has been the subject of several recent reviews [87,88,89,90,91,92], and it will thus not be reviewed here. However, it may be useful to shine light, in the context of the present review, on this “dark side of SOX2” [90]; it is possible that this emerges by hijacking the SOX2-driven gene regulatory network for the benefit of a pathological cell, instead of normal neurogenic stem cells. It is possible that aspects of the SOX2-driven “stem cell program” in normal and pathogenic (cancer) neural stem cells relies on similar target genes; on the other hand, some SOX2 targets may also differ in normal and tumorigenic stem cells, for example, via the presence of different SOX2 co-factors (see above, Section 4). Experimental manipulation of SOX2 targets in neural CSC/TIC will be important to ascertain their role, and potential usefulness from a therapy perspective. Indeed, SOX2 itself [91], as well as its targets (including the Notch pathway, transcription factors, signaling molecules [87,93]), are actively investigated from the perspective of inactivating the SOX2-controlled gene regulatory network in CSC/TIC, as a therapeutic perspective for neural, and other tumor types [87,88,90,91,92,93]. Various strategies aimed at targeting SOX2 itself have been proposed, ranging from raising immune response directed against SOX2, to interference with SOX2 binding to DNA, to small molecule inhibitors of the signaling pathways affecting SOX2, to the induction of SOX2 degradation (see [91], Figure 7). In addition, the structure of SOX2 interacting with importin, responsible for its translocation to the nucleus, was recently resolved, and amino acids important for SOX2-importin interaction identified; mutations interfering with SOX2-importin interaction also interfered with SOX2 function [94]. Hence, interfering with SOX2 nuclear translocation by small molecules able to antagonize SOX2-importin interaction may provide an additional therapy perspective for glioblastoma, and possibly other SOX2-dependent tumors.

## 7. Conclusions and Final Remarks

Mouse *Sox2* cKOs have been instrumental in understanding SOX2′s role in brain development and in shedding light on the causes of the defects found in human patients with mutations in the *SOX2* gene. Gene regulatory networks downstream of SOX2 in NSC have been identified, and some key SOX2 effectors in specific neurons and glia have been found.

Novel genomic techniques, such as spatial transcriptomics and single cell RNAseq, will allow us to determine in which specific cell types SOX2 loss has profound effects and how gene expression is affected. We have started to understand how some key SOX2 targets function in NSC, such as SOCS3/FOS/JUN, and it will be exciting in the future to study the role of SOX2 targets in neurons and glia.

## Figures and Tables

**Figure 1 cells-11-01604-f001:**
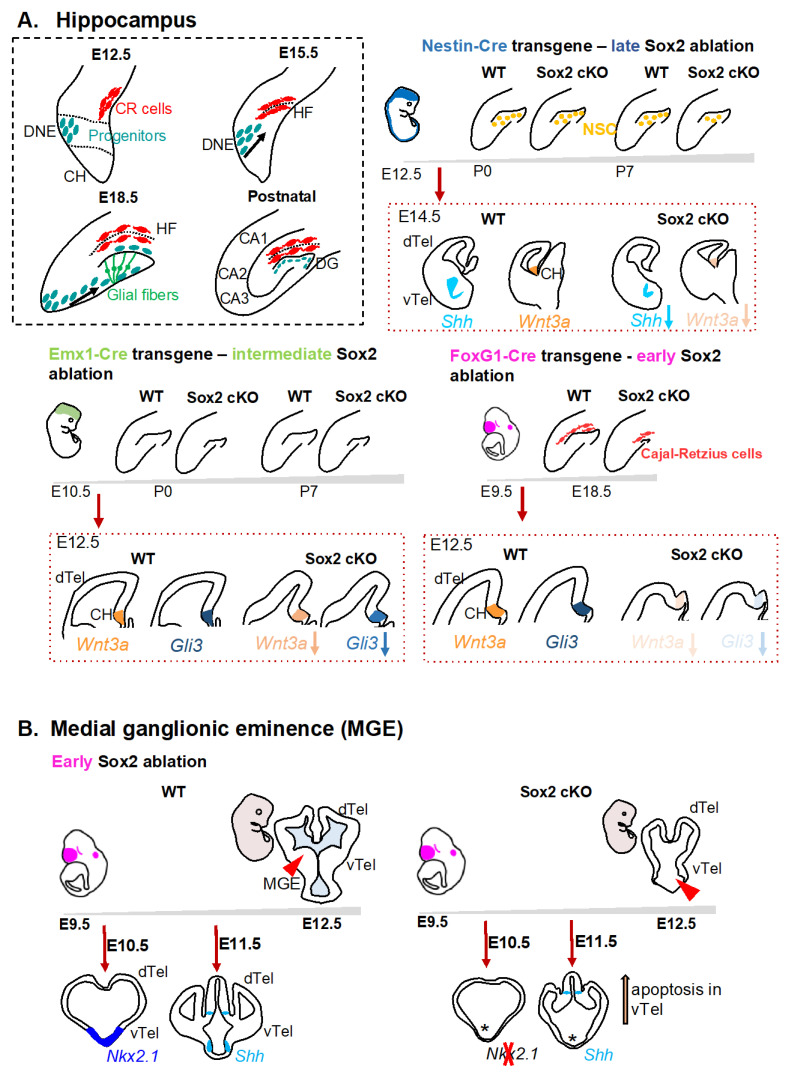
Schematic representation of defects arising from *Sox2* deletion in the telencephalon of *Sox2* cKO mice. (**A**). Hippocampus. Top left, schematic representation illustrating hippocampus development. Progenitor cells initially localized to the dentate neuroepithelium (DNE) migrate towards the forming dentate gyrus (DG) as the hippocampal fissure (HF) forms, Cajal-Retzius (CR) cells migrate to the HF, and the glial scaffold is organized. Postnatally the hippocampus main components are shown: cornu ammonis (CA) 1, CA2, CA3, and DG. Top right, “late” *Sox2* ablation, mediated by Nestin-Cre, leads to *Sox2* deletion by E14.5 in the whole nervous system and results in a reduction of NSC in the DG by postnatal day 7 (P7). At E14.5, expression of the secreted molecules SHH and WNT3a is reduced in the *Sox2* mutants in the ventral and medial telencephalon, respectively. Bottom left, “intermediate” *Sox2* ablation, mediated by Emx1-Cre, leads to *Sox2* deletion by E10.5 in the dorsal telencephalon and results in a strong reduction of NSC in the DG already at P0. Expression of key regulators of hippocampal development was analyzed and, while *Gli3* expression is not affected in Sox2 cKO, *Wnt3A* expression is slightly reduced. Bottom right, “early” *Sox2* ablation, mediated by FoxG1-Cre, leads to *Sox2* deletion by E9.5 in the whole telencephalon and results in greatly hypomorphic DG and reduced numbers of CR cells. Expression of the key regulators of hippocampal development, *Gli3* and *Wnt3A*, is greatly reduced in this *Sox2* cKO. (**B**). Medial ganglionic eminence (MGE). “Early” *Sox2* ablation, mediated by FoxG1-Cre, results in downregulation of expression of the ventral markers *Nkx2* and *Shh* and lack of the MGE in the Sox2 cKO compared to control siblings. Apoptosis in the MGE precedes the morphological defect.

**Figure 2 cells-11-01604-f002:**
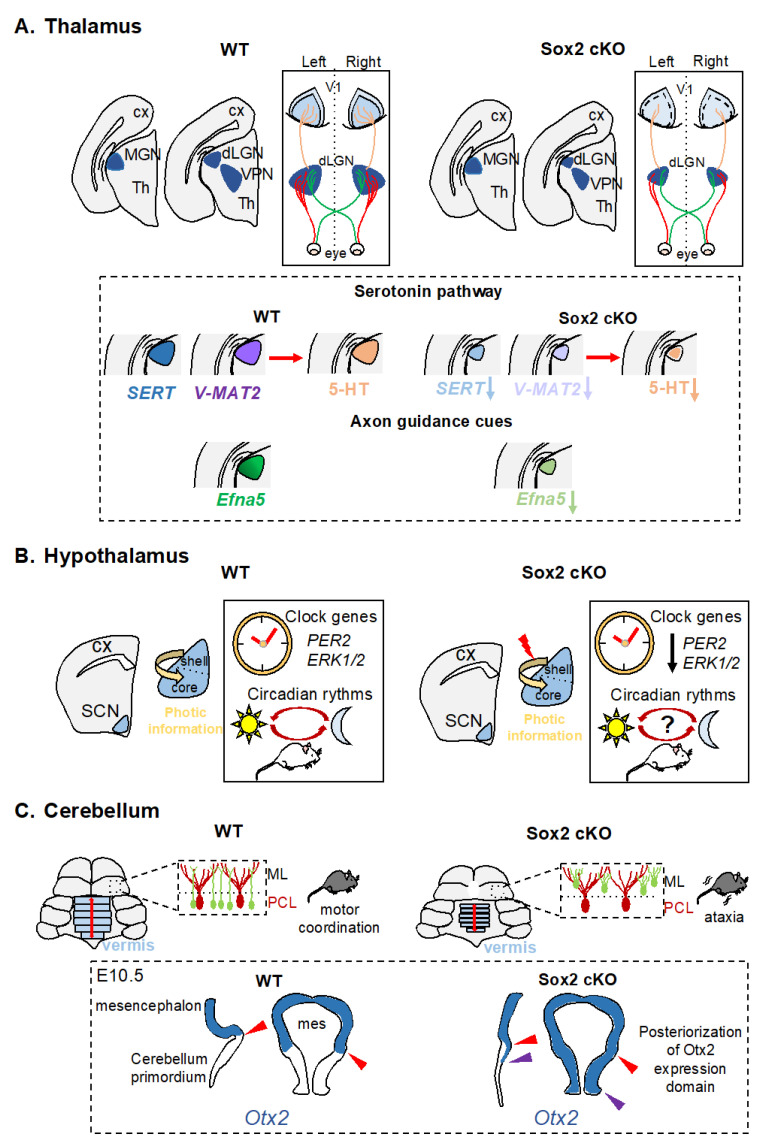
Schematic representation of defects arising from *Sox2* deletion in the thalamus, hypothalamus, and cerebellum of *Sox2* cKO mice. (**A**) Thalamus. *Sox2* thalamic KO, mediated by Roralpha-Cre, leads to ablation of *Sox2* expression in the medial geniculate nucleus (MGN, auditory), the dorsolateral geniculate nucleus (dLGN, visual) and the ventroposterior nucleus (VPN, somatosensory). A reduction of the dLGN is observed in *Sox2* cKO mice compared to controls (WT) and projections from the mutant dLGN to the primary visual cortex (V1) are reduced, leading to a mis-patterned V1. The expression of components of the Serotonin pathway and of the axon guidance molecule Ephrin A5 (Efna5) is reduced in *Sox2* thalamic mutants compared to controls. (**B**) Hypothalamus. *Sox2* conditional deletions by VGAT-Cre in clock neurons in the suprachiasmatic nucleus (SCN) disrupts circadian rhythms and results in downregulation of clock genes (*Per2* and *Erk1/2*). (**C**) Cerebellum. *Sox2* deletion in the cerebellum, mediated by Wnt1-Cre, leads to a reduction in the size of the cerebellar vermis, aberrant morphology, and misplacement of bergmann glia (BG) and motor control problems. The border of expression of *Otx2*, between midbrain and hindbrain, is moved posteriorly in *Sox2* cKO compared to controls. cx, cortex; th, thalamus; mes, mesencephalon.

**Figure 3 cells-11-01604-f003:**
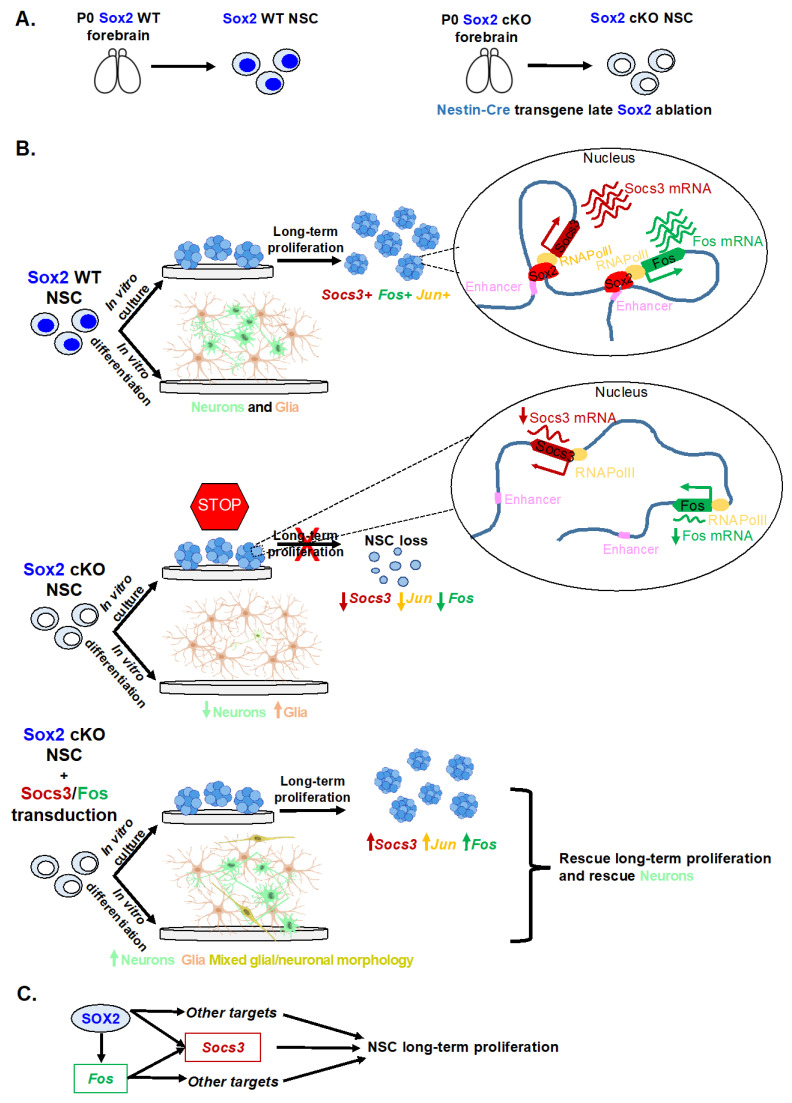
Schematic representation of the gene regulatory network downstream of SOX2 in forebrain-derived NSC, required for NSC proliferation and neuronal differentiation. (**A**) *Sox2*-deleted (Sox2 cKO) or wild-type (Sox2 WT) NSC were derived from the forebrain (at P0) of Sox2-Nestin-Cre cKO (“late” *Sox2* ablation in Figure 1) or control siblings, respectively. (**B**) Top panel, Sox2 WT NSC proliferate in long-term culture forming neurospheres and can differentiate into both neurons (green) and glia (brown). SOX2 binds to an enhancer and the promoter of both *Socs3* and *Fos* and activates their transcription. In turn, SOX2 and FOS (AP1) bind together to the *Socs3* promoter (and to many shared target genes genome-wide). Middle panel, NSC lacking *Sox2* (Sox2 cKO NSC) are unable to proliferate long-term in culture, and their ability to differentiate into neurons is compromised. In addition, the expression of *Socs3* and *Fos* is greatly reduced. Bottom panel, the overexpression of *Socs3* or *Fos*, via viral transduction, in *Sox2* cKO NSC rescues both long-term proliferation in culture and differentiation into neurons. (**C**) Regulatory relations between SOX2, FOS, and SOCS3, in neural stem cell (NSC) long-term self-renewal control.

**Table 1 cells-11-01604-t001:** Timing and location of *Sox2* deletion in different *Sox2* conditional knock-outs (cKO).

Sox2 cKO	Cre Expression Domain in CNS	Timing Sox2 Deletion	Reference
Sox2-FoxG1-Cre cKO	Telencephalon	complete by E9.5	[11,12]
Sox2-Emx1-Cre cKO	Dorsal Telencephalon	complete by E10.5	[12,13]
Sox2-Nestin-Cre cKO	NSC	complete by E14.5	[12,14,15]
Sox2-Wnt1-Cre cKO	Midbrain and Hindbrain	complete by E9.5	[16,17]
Sox2-Rora-Cre cKO	Thalamus (dLGN, VP, MG)	complete by E15.5	[18,19]

## Data Availability

Not applicable.

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
