# Peer review of "Deconstructing Sox2 Function in Brain Development and Disease"

_cells, 2022, doi:10.3390/cells11101604_

Round 1

Reviewer 1 Report

In their review article, Mercurio et al details how SOX2 affects neurologic development using mouse models as an experimental model organism, and how deficiency of SOX2 in various neural lineages or developmental timepoints differentially affects brain development. They further identify correlates between mouse and human development and link SOX2 mutations with human neurological disorders. The authors further explore possible mechanisms underlying tissue and developmental stage specificity for SOX2 function, including interactions with other developmental factors. The review is well written and informative and would be interesting to a broad audience. The manuscript seems similar to a previously published report by the same group:

Mercurio, S.; Serra, L.; Nicolis, S.K. More than just Stem Cells: Functional Roles of the Transcription Factor Sox2 in Differentiated Glia and Neurons. Int. J. Mol. Sci. 201920, 4540. https://doi.org/10.3390/ijms20184540

  1. The “telencephalon” and “diencephalon” sections appear to be much shorter than the other sections. Could the authors expand this discussion to include additional studies or explain if less is known about the roles of SOX2 in these anatomic regions?

  1. Many sentences throughout the review are very long and could be shortened or broken up into several sentences to facilitate understanding.

  1. The authors may consider briefly commenting on the roles of other SOX family members in nervous system development and contrast these roles with those of SOX2.

  1. It would be helpful to have a figure or table that illustrates the tissue and developmental timing specificities of various drivers used in the Cre recombinase system (such as FOXG1, Nestin, Emx1 etc).

  1. Figure 1: instead of using shading to illustrate factor expression level, it would be better to use arrows indicating “high” or “low” expression. It is difficult to appreciate differences in shading.

  1. Some text appears to be copied and pasted directly from published articles (lines 410, 415-416). The authors should not copy text word-for-word, but should instead summarize the main findings.

  1. The authors may wish to mention that SOX2 is one of the 4 critical factors involved in induced pluripotent stem cell induction and discuss its important role in stemness and pluripotency.

  1. The authors may wish to add additional discussion of the role of SOX2 in neoplastic brain diseases (such as glioblastomas or other brain tumors), as there are direct parallels between work on NSCs in neurodevelopment and glioblastoma stem cell populations. There is a rich literature on SOX2 in the brain tumor space, some of which is included here:

Zhu Z, Mesci P, Bernatchez JA, Gimple RC, Wang X, Schafer ST, Wettersten HI, Beck S, Clark AE, Wu Q, Prager BC, Kim LJY, Dhanwani R, Sharma S, Garancher A, Weis SM, Mack SC, Negraes PD, Trujillo CA, Penalva LO, Feng J, Lan Z, Zhang R, Wessel AW, Dhawan S, Diamond MS, Chen CC, Wechsler-Reya RJ, Gage FH, Hu H, Siqueira-Neto JL, Muotri AR, Cheresh DA, Rich JN. Zika Virus Targets Glioblastoma Stem Cells through a SOX2-Integrin αvβ5 Axis. Cell Stem Cell. 2020 Feb 6;26(2):187-204.e10. doi: 10.1016/j.stem.2019.11.016. Epub 2020 Jan 16. PMID: 31956038.

Hubert CG, Rivera M, Spangler LC, Wu Q, Mack SC, Prager BC, Couce M, McLendon RE, Sloan AE, Rich JN. A Three-Dimensional Organoid Culture System Derived from Human Glioblastomas Recapitulates the Hypoxic Gradients and Cancer Stem Cell Heterogeneity of Tumors Found In Vivo. Cancer Res. 2016 Apr 15;76(8):2465-77. doi: 10.1158/0008-5472.CAN-15-2402. Epub 2016 Feb 19. PMID: 26896279; PMCID: PMC4873351.

Suvà ML, Rheinbay E, Gillespie SM, Patel AP, Wakimoto H, Rabkin SD, Riggi N, Chi AS, Cahill DP, Nahed BV, Curry WT, Martuza RL, Rivera MN, Rossetti N, Kasif S, Beik S, Kadri S, Tirosh I, Wortman I, Shalek AK, Rozenblatt-Rosen O, Regev A, Louis DN, Bernstein BE. Reconstructing and reprogramming the tumor-propagating potential of glioblastoma stem-like cells. Cell. 2014 Apr 24;157(3):580-94. doi: 10.1016/j.cell.2014.02.030. Epub 2014 Apr 10. PMID: 24726434; PMCID: PMC4004670.

Bulstrode H, Johnstone E, Marques-Torrejon MA, Ferguson KM, Bressan RB, Blin C, Grant V, Gogolok S, Gangoso E, Gagrica S, Ender C, Fotaki V, Sproul D, Bertone P, Pollard SM. Elevated FOXG1 and SOX2 in glioblastoma enforces neural stem cell identity through transcriptional control of cell cycle and epigenetic regulators. Genes Dev. 2017 Apr 15;31(8):757-773. doi: 10.1101/gad.293027.116. Epub 2017 May 2. PMID: 28465359; PMCID: PMC5435889.

Ahlfeld J, Favaro R, Pagella P, Kretzschmar HA, Nicolis S, Schüller U. Sox2 requirement in sonic hedgehog-associated medulloblastoma. Cancer Res. 2013 Jun 15;73(12):3796-807. doi: 10.1158/0008-5472.CAN-13-0238. Epub 2013 Apr 17. PMID: 23596255.

Tao R, Murad N, Xu Z, Zhang P, Okonechnikov K, Kool M, Rivero-Hinojosa S, Lazarski C, Zheng P, Liu Y, Eberhart CG, Rood BR, Packer R, Pei Y. MYC Drives Group 3 Medulloblastoma through Transformation of Sox2+ Astrocyte Progenitor Cells. Cancer Res. 2019 Apr 15;79(8):1967-1980. doi: 10.1158/0008-5472.CAN-18-1787. Epub 2019 Mar 12. PMID: 30862721; PMCID: PMC6467710.

Lopez-Bertoni, H., Johnson, A., Rui, Y. et al. Sox2 induces glioblastoma cell stemness and tumor propagation by repressing TET2 and deregulating 5hmC and 5mC DNA modifications. Sig Transduct Target Ther 7, 37 (2022). https://doi.org/10.1038/s41392-021-00857-0

Riddick, G., Kotliarova, S., Rodriguez, V. et al. A Core Regulatory Circuit in Glioblastoma Stem Cells Links MAPK Activation to a Transcriptional Program of Neural Stem Cell Identity. Sci Rep 7, 43605 (2017). https://doi.org/10.1038/srep43605

Stevanovic M, Kovacevic-Grujicic N, Mojsin M, Milivojevic M, Drakulic D. SOX transcription factors and glioma stem cells: Choosing between stemness and differentiation. World J Stem Cells. 2021;13(10):1417-1445. doi:10.4252/wjsc.v13.i10.1417

  1. Could consider citing:

Sikorska M, Sandhu JK, Deb-Rinker P, Jezierski A, Leblanc J, Charlebois C, Ribecco-Lutkiewicz M, Bani-Yaghoub M, Walker PR. Epigenetic modifications of SOX2 enhancers, SRR1 and SRR2, correlate with in vitro neural differentiation. J Neurosci Res. 2008 Jun;86(8):1680-93. doi: 10.1002/jnr.21635. PMID: 18293417.

Morgado AL, Rodrigues CM, Solá S. MicroRNA-145 Regulates Neural Stem Cell Differentiation Through the Sox2-Lin28/let-7 Signaling Pathway. Stem Cells. 2016 May;34(5):1386-95. doi: 10.1002/stem.2309. Epub 2016 Feb 29. PMID: 26849971.

Zhang Y, Kim MS, Jia B, Yan J, Zuniga-Hertz JP, Han C, Cai D. Hypothalamic stem cells control ageing speed partly through exosomal miRNAs. Nature. 2017 Aug 3;548(7665):52-57. doi: 10.1038/nature23282. Epub 2017 Jul 26. Erratum in: Nature. 2018 Aug;560(7719):E33. PMID: 28746310; PMCID: PMC5999038.

Julian LM, Vandenbosch R, Pakenham CA, Andrusiak MG, Nguyen AP, McClellan KA, Svoboda DS, Lagace DC, Park DS, Leone G, Blais A, Slack RS. Opposing regulation of Sox2 by cell-cycle effectors E2f3a and E2f3b in neural stem cells. Cell Stem Cell. 2013 Apr 4;12(4):440-52. doi: 10.1016/j.stem.2013.02.001. Epub 2013 Mar 14. PMID: 23499385.

Julian LM, Vandenbosch R, Pakenham CA, Andrusiak MG, Nguyen AP, McClellan KA, Svoboda DS, Lagace DC, Park DS, Leone G, Blais A, Slack RS. Opposing regulation of Sox2 by cell-cycle effectors E2f3a and E2f3b in neural stem cells. Cell Stem Cell. 2013 Apr 4;12(4):440-52. doi: 10.1016/j.stem.2013.02.001. Epub 2013 Mar 14. PMID: 23499385.

Liu YR, Laghari ZA, Novoa CA, Hughes J, Webster JR, Goodwin PE, Wheatley SP, Scotting PJ. Sox2 acts as a transcriptional repressor in neural stem cells. BMC Neurosci. 2014 Aug 8;15:95. doi: 10.1186/1471-2202-15-95. PMID: 25103589; PMCID: PMC4148960.

Andreu-Agullo C, Maurin T, Thompson CB, Lai EC. Ars2 maintains neural stem-cell identity through direct transcriptional activation of Sox2. Nature. 2011 Dec 25;481(7380):195-8. doi: 10.1038/nature10712. PMID: 22198669; PMCID: PMC3261657.

Cui CP, Zhang Y, Wang C, Yuan F, Li H, Yao Y, Chen Y, Li C, Wei W, Liu CH, He F, Liu Y, Zhang L. Dynamic ubiquitylation of Sox2 regulates proteostasis and governs neural progenitor cell differentiation. Nat Commun. 2018 Nov 7;9(1):4648. doi: 10.1038/s41467-018-07025-z. Erratum in: Nat Commun. 2019 Jan 8;10(1):173. PMID: 30405104; PMCID: PMC6220269.

Feng R, Zhou S, Liu Y, Song D, Luan Z, Dai X, Li Y, Tang N, Wen J, Li L. Sox2 protects neural stem cells from apoptosis via up-regulating survivin expression. Biochem J. 2013 Mar 15;450(3):459-68. doi: 10.1042/BJ20120924. PMID: 23301561.

Reviewer 2 Report

This review addresses the role of Sox2 during nervous system development. The review is well-written and timely, and covers all of the main areas of research where Sox2 has been studied. I think it will be an important and valuable review for the many researchers in the areas of developmental biology, stem cell biology and CNS disorders. I have only a few comments for how to possibly improve the piece.

Major:

  • During development, in most CNS regions Sox2 is expressed at high levels in progenitors and at lower levels in various daughter cells, and sometimes in neurons and/or glia. These differences in Sox2 levels are not commented upon, rather Sox2 is mostly referred to as “expressed”. Perhaps, the levels of Sox2 expression could be discussed, as they are likely to play a functional role.

  • Rows 427-431: Please clarify if the heterozygotic mutations in humans leading to anophtalmia are predicted to be loss-of-function, gain-of-function or neomorphic. The Sox2 gene is not predicted to be haploinsufficient by GnomAD, so I would gather that these mutants may be GOF or neomorphic?

  • Section 3, rows 325-384. In the section on the role of Sox2 in NSCs they do not comment on the fact that Sox2 is also one of the original members identified for iPSC reprogramming. How does the role of Sox2 in NSCs, and I guess CNS, differ from its role in iPSC, and I guess early embryo? I am not sure this is well understood, and may therefore be difficult to elaborate on, but I nevertheless think that this should be briefly discussed.

Minor:

  • Row 48: Suggest change “We will show...” to “We will review findings that...”, or something similar. “We will show…” sounds as if the authors are conducting experiments, when they are in fact only reviewing previous findings.

  • Rows 229-230: gap in the text.

  • Rows 378 and 384: They reference their own unpublished data. Not sure if this is allowed in this journal.

  • Row 405, 411, 423: spaces.

  • Rows 410-11, 414-417: different font.

Reviewer 3 Report

In this, the authors provide an overview about Sox2 expression and function in different parts of the brain, mechanistic insights and suggestions, as well as disease related aspects.

This review is logically conceptualised, well balanced, and different aspects are supported by helpful schematic illustrations.

As these schemes are very helpful, I would advice to include also the aspect, mentioned in line 76ff: „„NSC and intermediate neural progenitors (IP) migrate from the dentate neural epithelium (DNE), adjacent to the CH, along glial fibers to eventually organize the DG. Cajal-Retzius cells, derived from the CH, have a key role in DG formation [10,13,14]}. „ 

in a scheme, as this might not be easy to understand for readers that are not specialists in hippocampal development.

In matters of cortical interneurons (Line 139ff), please be more specific: „Cortical interneurons (CIN) constitute are inhibitory neurons in the cerebral cortex 139 and they are essential in regulating communication between cortical neurons [23,24]. „

Of course, there are also excitatory interneurons in the cortex (spiny stellate cells), so please specify (gamma amino butyric acid (GABA) positive interneurons), and introduce GABA here 

Otherwise, I do not have major remarks. I would strongly advise to revise the manuscript in matters of errors and spelling (e.g. line 404: function; Line 139: „Cortical interneurons (CIN) constitute are …“), grammar and especially gene/protein nomenclature. Protein symbols are in capital letters, and gene symbols should be in italic (and in capital letters in case of human genes). 

The abstract frequently contains „defects“, which should be improved.
